# Environmental disclosure practices in mixed ownership models: A study of Chinese private enterprises

**Tingting Song** [ID]*, **Aihua Xiong**

Department of Business Administration, Shandong University of Finance and Economics, Jinan, China

* 211107003@mail.sdufe.edu.cn

**Data Availability Statement:** Data were obtained from the CSMAR database, which can be obtained from https://data.csmar.com/.

**Funding:** The author(s) received no specific funding for this work.

## Abstract

Environmental information disclosure is critical avenue for stakeholders to gauge the fulfillment of corporate environmental responsibilities, as well as a key path for companies to gain social reputation and achieve sustainable development. To achieve both economic and social sustainability and improve the environmental information disclosure by private firms, this study delves into the impact mechanism and realization path of mixed ownership reform on environmental information disclosure among Chinese private enterprises listed between 2010 and 2020. Utilizing a panel fixed effect model, we scrutinize the interplay between state capital involvement and the disclosure of environmental information by private enterprises. Our findings reveal that state capital involvement may encourage private enterprises to disclose environmental information through resource allocation and governance improvements. The higher the shareholding ratio of state-owned participating shareholders, the more it helps private firms to disclose environmental information. State-owned shareholders play a pivotal role in the appointment of supervisors, directors, and executives, effectively improving corporate governance mechanisms and positively moderates how private companies with state-owned capital participate in disclosing environmental information. Moreover, the magnitude of media coverage and the public opinion pressure faced by private enterprises further amplify the influence of state-owned capital involvement on environmental information disclosure. Additionally, our research reveals that corporate profitability partially interplays with the effects of state-owned capital disclosure of environmental information by private companies. According to the research results, we recommend that the government proactively promote mixed ownership reform with private enterprises as the main participants, fully leveraging the resource advantages and influence of state-owned capital. At the same time, it is imperative to strengthen the governance effect of internal state-owned shareholders and external public opinion supervision in private enterprises. Enhancing profitability is also identified as a key driver for private enterprises to engage in more robust environmental information disclosure practices.

**Competing interests:** The authors have declared that no competing interests exist.

## 1. Introduction

Environmental pollution is growing increasingly serious as the social economy develops rapidly. Sustainable development, low carbon, and environmental protection have become a common desire for all countries and regions worldwide. Enterprises are important social development promoters and major contributors to environmental and ecological pollution. Enterprises should take primary responsibility for pollution management and environmental protection [1]. The corporate environmental information disclosure (EID) has emerged as a vital instrument for environmental governance, garnering increasing attention from governments and the public worldwide [2]. During the last two decades, China has made great economic achievements but also confronted environmental challenges [3]. In response, the Chinese government has made some attempts to increase corporate environmental consciousness and encourage businesses to proactively publish environmental information to meet the environmental requirements. The State Environmental Protection Administration (SEPA) issued Chinese first regulatory document on corporate EID in 2003, the Announcement on Corporate EID. This announcement mandated heavily polluting enterprises that exceeded emission standards in disclosing five aspects of corporate environmental policies and pollutant emissions, and encouraged voluntary disclosure of additional environmental information. However, it did not explicitly define the format of disclosure, nor did it stipulate fines for noncompliance. In 2008, the SEPA promulgated the Measures for EID for Trial Implementation, emphasizing the timely and accurate disclosure of environmental information by enterprises while complying with the concept of combining voluntary and obligatory disclosure. It also added incentives for environmentally responsible companies, including public recognition and priority arrangement of special funding projects. Environmental and Ecological Protection: The Thirteenth Five-Year Plan in 2016, announced by the State Council, recommended establishing an obligatory disclosure method for environmental protection information and penalizing businesses that fail to comply with disclosure duties. In 2020, the Central Committee for Comprehensively Deepening Reform held its 17th meeting to discuss and approve the Reform Plan for the Environmental Information Disclosure Law. This marked a significant step toward creating comprehensive environmental data disclosure requirements in China, characterized by self-discipline, effective management, strict supervision, and strong enterprise support. Measures for the Management of Business Exposure of Environmental Data According to Law were published by the Ministry of Environment and Conservation in 2021, which outlines the legal requirements for corporate EID.

These measures underscore China's growing commitment to EID, positively influencing the EID practices of listed firms. However, compared to developed countries, EID in China remains in its nascent stage, primarily due to the lack of robust laws, regulations, and punishment mechanisms for corporate EID. This deficiency is particularly pronounced among private enterprises [4]. Zeng et al. [5] investigated the EID status of China's listed manufacturing enterprises and discovered that the EID was influenced by the nature of ownership of the majority shareholder, industry, and company size, with commercial companies having lower EID rates than state-owned counterparts. In the Chinese context, EID of listed companies is mostly not mandatory and reflecting their sense of social responsibility. Compared to state-owned enterprises, private enterprises do not have excessive policy burdens. Some private enterprises often disclose environmental information based on strategic needs [6], with varying degree of awareness regarding self-disclosure of environmental information. In addition, environmental governance is marked by high investments, high risks, and long timeframes, making it difficult to achieve results in a short period of time [7]. Private enterprises often face resource acquisition limitations [8], resulting in insufficient investments in environmental

governance making it difficult to improve the level and quality of EID. Private businesses are a key component of China's economic and social development and are essential for advancing high-quality sustainable economic growth and environmental protection. Thus, exploring the determinants of EID by private enterprises within the Chinese context has become a major issue that necessitates collaboration among the government, market, and corporate entities.

Recently, China has been actively implementing changes to facilitate mixed ownership. Microeconomic mixed ownership involves intertwining state-owned, collective, and non-public capital through mutual integration and cross-shareholding. Hence, private funds may invest in state-owned enterprises, or state-owned capital may invest in private companies (including private firms) [9]. State-owned capital participation in private enterprises carries a two-fold impact. On one hand, the equity affiliation of private firms with state-owned shareholders strengthen ties with the government and may facilitate access to resources and reputational capital, thereby enhancing potential capital investment in environmental governance. On the other hand, state-owned shareholders, driven by political needs and social norms, may exert pressure on private firms for environmental governance. Concurrently, they convey a sense of environmental awareness, thereby increasing the likelihood that private firms will partake in EID. Therefore, it is urgent for us to further confirm whether mixed ownership reform can promote private enterprises to actively disclose environmental information.

Using a sample of privately listed Chinese companies from 2010 to 2020, this study empirically examines the effect of state ownership of capital on the EID. This research contributes to the field in two significant ways. Firstly, while previous literature has explored the internal factors that affect corporate EID in China, mainly focusing on company characteristics [10], executive characteristics [11], shareholder pressure [12] etc. Few scholars have studied the impact of equity structure on corporate EID. Moreover, previous research on the equity structure of enterprises has predominantly focused on a single dimension of state-owned and non-state-owned equity [13, 14]. This article examines the level of EID in private enterprises under mixed equity in the Chinese context, demonstrating that mixed equity can effectively integrate resources and improve governance mechanisms, thereby altering the EID decisions of enterprises, and to some extent, enriching literature research on enterprise environmental information disclosure. Secondly, existing literature on mixed ownership reform mainly focuses on state-owned enterprises and explores the consequences of private capital entering state-owned enterprises, such as corporate performance [15], corporate innovation [8], excessive perquisite consumption by executives [16], green transformation [17], with relatively limited attention given to mixed reforms involving private enterprises. A small number of scholars have explored the impact of private enterprises reform on corporate default risk [18] and social responsibility [19], etc. Surprisingly, the mechanism by which the disclosure of environmental information in private enterprises is influenced has remained unexplored. This paper takes private enterprises as the participating subject, analyzes the mechanism by which state-owned capital participation affects the EID of private firms from the perspectives of resource effects and governance effects, thus supplementing research on the path of state-owned capital participation to promote the EID of private firms.

## 2. Literature review

Current research in the field of EID has primarily focused on three key areas. Firstly, analyzing the correlation between environmental performance and environmental information disclosure has not yet reached a consistent conclusion. Acar and Temiz [20] supported a positive correlation between environmental performance and EID, and they argued that better environmental performers are more willing to participate in EID. Patten's study suggested a

negative relationship between environmental performance and EID, arguing that firms use environmental information as a legitimate tool, and that companies with poor environmental performance will disclose more positive environmental information [21]. Meng et al. [22] found a non-linear relationship between corporate environmental performance and EID through content analysis of 533 listed companies. Secondly, existing literature has explored the economic consequences of EID, including financial performance [23], green innovation [24], and stock price collapse risk [25]. The third aspect of research has been focused on the factors influencing EID. It mainly focuses on both external and internal aspects of enterprises. From an external perspective of enterprises, formal environmental regulatory policies force enterprises to comply with regulations in order to obtain legitimacy, resulting in a tendency to disclose high-quality environmental information [26]. Under the informal regulatory pressure of media supervision [27] and stakeholders [12], companies usually disclose environmental information in order to maintain their reputation. In addition, the degree of EID by enterprises may also vary due to policy uncertainty [28] and the level of industry and market competition [12]. From an internal perspective, factors such as firm characteristics [10], profitability [29], board characteristics [30, 31], and corporate governance [32] can all affect corporate EID to some extent.

The vast majority of academics have applied legitimacy theory to explain the disclosure practices of enterprises regarding environmental information, which holds that there is a contractual relationship between enterprises and society and that enterprises must seek recognition from society and the public for their legitimacy to sustain their business operations [33]. Specifically, concerning the environment, companies' production and operation will inevitably cause environmental pollution. Enterprises may voluntarily provide environmental information in response to growing public scrutiny to reassure consumers of their social obligation to safeguard the environment. Legitimacy theory treats information disclosure as a function of social and governmental pressure [34]. Enterprises maintain the legitimacy of their existence by communicating their environmental management behavior and performance through EID in dialogue with the social public to meet social demands for environmental legitimacy. In China, government pressure is the major external pressure source on corporate EID. Menguc et al. [35] established a substantial positive impact of government regulation of corporate information disclosure. In addition to formal mechanisms of influence, such as regulations and laws, the influence of informal mechanisms should not be ignored. Political correlations, a typical link in emerging nations [36], serves as a crucial informal channel between businesses and the government. However, scholars have produced varying research conclusions regarding the impact of political connections on EID. Cheng et al. [37] empirically analyzed the impact of political affiliations on companies' proactive disclosure of environmental information based on data from heavily polluting listed companies in China. Li et al. [38] believed that the political affiliation of executives is negatively correlated with companies' EID. Notably, after state-owned capital shares in private enterprises, private firms can benefit from strengthened partnerships with the government, thanks to their state-owned shareholders' political attributes. Unlike the political connection of senior executives, the equity connection with state-owned shareholders is formal, establishing a natural link between private firms and the government. As a state-owned capital closely related to the government, what impact will it have on the EID of private enterprises after participating in private enterprises? Does this equity linkage have a different outcome on corporate EID than the political linkage of corporate executives? Therefore, the impact of state-owned capital participation on environmental information disclosure in private enterprises deserves further research. Therefore, based on the theory of resource dependence and agency cost, this article explores the impact mechanism of the resource effect and governance effect of state-owned capital participation on the EID in

private enterprises. Insufficient literature exists that uses resource-based theory alone to examine what drives environmental information disclosure. Most scholars still use resource dependence theory as an auxiliary theory and combine it with legitimacy theory to study the influencing factors of corporate EID. For example, Branco and Rodrigues [39] point out that to obtain certain intangible resources, companies need to maintain good relationships with stakeholders by disclosing social responsibility information, including environmental information, to make it easier to maintain intangible assets. In this article, we examine the resource impact of state-owned companies on how government and businesses communicate about EID, making the resource dependence theory, a classical theory of organization-environment and organization-organization interaction, highly relevant to this study.

## 3. Theoretical analysis and research hypothesis

### 3.1 State-owned capital participation and environmental information disclosure

According to resource dependency theory, organizations must obtain the necessary resources for survival and sustainable development through the external environment, and these resources are not under their control, thus creating a dependency on the controller of the resources. In the Chinese context, the government holds key resources for corporate development, and due to the government background of state-owned enterprises, they possess certain advantages in accessing these crucial resources. In contrast, private enterprises suffer from property rights discrimination and constraints in resource availability [40]. It also limits the environmental governance investments of private enterprises to some extent, thereby affecting their EID. Cheng et al. [37] found that private firms are able to access key resources for corporate development and political reputation through political affiliation, which enhances the level of corporate EID. Li et al. [19] pointed out that state-owned capital participation can help private enterprises alleviate financing constraints and enhance political status, and foster corporate social responsibility commitments, thus incentivizing greater social responsibility on the part of enterprises. State-owned capital's involvement in private enterprises establishes an invisible and stable political connection with the government, leading to resource effects for private enterprises [41], such as breaking down industry barriers and gaining access to government subsidies and tax incentives. In addition, state-owned shareholders are an important reputational mechanism that can help private firms, attracting external investors [42], alleviating financing constraints, and enhancing economic and environmental performance, which, in turn, promotes EID.

Investments by state-owned entities in privately held companies can bring resource effects and potential governance effects. Principle-agent theory says that the principal usually cannot fully grasp the agent's decision-making behavior due to information asymmetry. Therefore, senior executives may make behavioral decisions inconsistent with shareholders' interests. Executives tend to maximize their interests and focus more on the enterprise's short-term benefits. However, environmental problems are persistent, and the cost of environmental protection is high; the self-interested behavior of executives may invest relatively little in environmental protection, resulting in poor environmental performance. Research has shown that companies with low environmental performance are less inclined to share reliable data on environmental information Acar and Temiz [20]. State-owned capital joining private enterprises, state-owned equity may play a role in equity balance, which can alleviate challenges associated with "insider control" and a lack of oversight caused by the controlling shareholder's "dominant share" to a certain extent. For example, the behaviors of controlling shareholders and management can be supervised, reducing collusion between them, and enhancing the

governance mechanisms in place [41]. Furthermore, state-owned capital has the quality of public policy and is instrumental in many areas, including public service provision and environmental protection. Its investment usually serves the national strategic objectives [41]. In addition to ensuring the preservation and appreciation of value, and improving the economic benefits of investment, it is imperative to consider social benefits. According to Zeng et al. [43], due to the intrinsic features of their ownership structure, state-owned enterprises are inherently motivated to provide environmental information than other types of organizations. The use of State-Owned Capital in the private sector may have political motives, making private enterprises more accountable for their social responsibilities [44] and subject to greater government supervision. While the government supports enterprises for development, also requiring them to fulfill their environmental responsibilities in line with national development goals. To establish political credibility, preserving this political relationship, and securing additional funding for development, businesses may become more active in disclosing environmental information in exchange for the government's assistance.

$H_1$: State-owned capital participation in private firms can promote their level of environmental information disclosure.

## 3.2 The moderating effect of media attention

Growing global environmental awareness has led to increased emphasis on public oversight, with the media playing a crucial role in information dissemination. The media has unique advantages in information dissemination. The media's function as an information intermediary in the financial market has grown significantly, particularly in digital age and the increasing importance of the Internet [45]. Kuo et al. [46] believed that the public and the government mainly understand the environmental management and implementation of enterprises through media coverage. As the world focuses on ecological civilization, the media closely monitors corporate environmental performance. Therefore, the media is an important monitoring force for corporate EID. Aert et al. [47] pointed out that firms with high media attention feel the pressure of public opinion. Kong et al. [48] verified the positive correlation between public opinion monitoring and corporate environmental communication in terms of the content of media coverage. The higher the number of environmental articles reported in the media related to the firm, the higher the quality and quantity of the firm's environmental disclosure. When media attention is low, enterprises that the state government owns (SOEs) and enterprises from private proprietors are under less pressure to monitor and have less influence on corporate EID. However, when media attention is significant, participating SOEs and private enterprises are subject to greater public opinion monitoring pressure. SOEs may exert more policy and compliance pressure on private enterprises to maintain the government's image. State-owned shareholders may also influence the EID decisions of enterprises through governance mechanisms for their own political performance goals and political futures, thus enhancing the level of EID of private enterprises.

$H_2$: Media attention positively moderates how state-owned equity involvement might encourage private companies to disclose environmental facts.

## 3.3 Moderating effect of executives, supervisors, and directors appointed by state-owned shareholders

The board of directors, which is at the center of corporate governance, plays a crucial role in addressing and resolving agency concerns and upholding the interests of all stakeholders. As

an integral component of the company's internal governance process, the board of supervisors may be in charge of monitoring directors and senior managers to perform their duties on behalf of shareholders. These supervisors, directors, and senior managers will be directly involved in the company's governance when state-owned enterprises funds are invested in a private company and board members, supervisors, and senior managers are chosen. Their role in overseeing directors, senior managers, and supervisors of private enterprises have the potential to improve corporate governance mechanisms, with their primary objective being the protection of shareholders' rights, economic efficiency, and the interests of private enterprises. Some studies have shown that environmental disclosure by enterprises helps to enhance enterprise value [49], promotes sustainable development, alleviates information asymmetry, and agency costs of enterprises, and safeguards the rights and interests of other shareholders. State-owned shareholders, by appointing directors, supervisors, and executives, assume the roles of checks and balances and supervision. To a certain extent, they can influence the EID decisions of enterprises. Due to their special political status, state-owned shareholders have a dual mission of politics and economy. For the sake of their political career, state-owned shareholders usually need to consider their political performance evaluation goals and align with the country's sustainable development strategy. The directors, supervisors, and senior management members appointed by state-owned shareholders will actively convey the importance of environmental protection to other senior managers, promote the rational and effective use of environmental resources, actively fulfill environmental responsibilities [50], and improve EID.

H₃: The appointment of senior executives, supervisors, and directors by state-owned shareholders positively tames the encouraging impact of state-owned equity participation on private companies' environmental data disclosure.

## 4. Methodology

### 4.1 Data

The listed private enterprises in Shenzhen and Shanghai from 2010 to 2020 were evaluated in this paper. The relevant financial data of these listed private enterprises, as well as the shareholding information of the top ten shareholders, particularly state-owned capital participation data, are sourced from China Stock Market & Accounting Research (CSMAR) Database, a widely used database for Chinese listed firms. EID data was manually collected from reports generated by businesses on their activities related to environmental responsibility, social responsibility, corporate social responsibility, and general operations. On this basis, the following processing and screening procedures are performed: (1) exclusion of companies marked as ST, *ST, and others with obvious abnormal financial status; (2) exclusion of financial-listed enterprises; (3) exclusion of enterprises with missing data. After these processing steps, the final sample of 14339 observations of listed private enterprises was obtained. To account for the potential influence of extreme values on regression results, this paper has carried out the Winsorize treatment at the 1% and 99% levels for all continuous variables of the sample data of enterprises.

### 4.2 Model

Considering the data characteristics and to test hypotheses H₁-H₃, we used the fixed effects model to test the relationship between state-owned capital participation and EID of private enterprises:

$$EID_{it} = \alpha_0 + \alpha_1 State_{it} + \sum Control-Variables + Year + Industry + \varepsilon_{it} \qquad (1)$$

$$EID_{it} = \alpha_0 + \alpha_1 State_{it} + \alpha_2 Media_{it} + \alpha_3 State_{it} \times Media_{it} + \sum Control-Variables + Year$$
$$+ Industry + \varepsilon_{it} \tag{2}$$

$$EID_{it} = \alpha_0 + \alpha_1 State_{it} + \alpha_2 Govs_{it} + \alpha_3 State_{it} \times Govs_{it} + \sum Control-Variables + Year$$
$$+ Industry + \varepsilon_{it} \tag{3}$$

Model (1) tests the impact of equity structure on the EID of private enterprises. Model (2) was used to examine the effect of press coverage on the connection between state-owned capital and EID at private firms. Model (3) examines the effect of state-owned shareholders' nomination of supervisors, directors, and executives on the correlation between the EID of private enterprises and state-owned capital involvement, where 0 is a constant term. The Control-Variables are a collection of control parameters, firms, and years indicated by the subscripts i and t, and $\varepsilon_{it}$ denotes a random disturbance term.

## 4.3 Variable measurement

**Explained variables (environmental information disclosure (EID)).**   We referred to the study by Chen et al. [10] using the "content analysis method" to quantify the EID of enterprises. We divided the environmental information disclosed by the company into six parts, including disclosure vehicles, environmental management, environmental costs and inputs, environmental liabilities, environmental governance and performance, governmental regulation, and institutional certification. The rules are as follows: A score of zero means there was no information disclosed, while a score of 1 signifies the presence of information. For environmental liabilities and environmental governance and performance, zero means no description, 1 represents a general qualitative description, and a score of 2 indicates a quantitative description. These six indicators are scored and summed to obtain each sample firm's total EID score (EIDI) and to reflect the degree of EID of different sample firms. First, we calculated the maximum score of 29 points for companies with respect to EID (The specific assignment of indicators is shown in Table 1) and then divided sample firms' EID scores by the maximum score of EID to obtain the company's EID index. The calculation formula is EID = 100* (EIDI/29).

**Explanatory variables (state-owned capital participation (state)).**   In this study, we characterized the explanatory variables using the state-owned participating shareholders' overall shareholding percentage among the top ten shareholders of private enterprises. This characterization aligns with prior research.

**Moderating variables (media attention).**   This variable was measured as the logarithmic value of the sum of the number of reports on the target company by online media and traditional paper media plus one. We use the Python programming language to conclude the statistics of the number of media reports, ensuring the reliability of the data. We conducted the data collection for online media reports using the Baidu search engine, which holds a substantial market share. Hence, this paper uses the enterprise news data retrieved from the Baidu engine. For traditional paper media reports, we relied on the eight most influential financial newspapers as the sources of reports. Senior governance was measured using the size of supervisors, directors, and executives appointed by shareholders owned by the state and by going through annual reports. If a director, supervisor, or senior member serves in a state-owned shareholder unit, the member is considered to be appointed by the state-owned shareholder.

**Control variables.**   This paper includes control variables aligned with existing literature, focusing on firm characteristics: enterprise size (Size), enterprise age (Age), chairman and

**Table 1. Corporate EID indicators.**

| Level 1 Indicators | Secondary indicators | Numerical value | | |
|---|---|---|---|---|
| | | Undisclosed | Qualitative Disclosure | Quantitative Disclosure |
| Environmental Liabilities | Discharge of wastewater | 0 | 1 | 2 |
| | Emissions of $SO_2$ | 0 | 1 | 2 |
| | Emissions of $CO_2$ | 0 | 1 | 2 |
| | Emissions of smoke and dust | 0 | 1 | 2 |
| | Generation of solid waste in industry | 0 | 1 | 2 |
| Environmental Governance and Performance | Techniques for lowering emissions of exhaust gas | 0 | 1 | 2 |
| | Wastewater abatement treatment | 0 | 1 | 2 |
| | Cleaner production implementation | 0 | 1 | 2 |
| | Use and disposal of solid waste. | 0 | 1 | 2 |
| Disclosure Vehicle | List Companies' Annual Reports | | | |
| | Social Responsibility Report | | | |
| Environmental Costs and Inputs | Environmental expenses | | | |
| | Total environmental investment | | | |
| Environmental Management | Training and education in environmental studies | Have for 1, not have for 0 | | |
| | The environmental incident response mechanism | | | |
| | Environmental concept | | | |
| | Environmental goals | | | |
| Government Regulation and Institutional Certification | The implementation of the " triple concurrent" system | | | |
| | ISO environmental certification implementation | | | |
| | Environmental honors | | | |

CEO concurrently (Dual), cash flow (Cash), board size (Boards), market competition (Com), percentage of independent directors (Indep), financial leverage (Lev), ratio of the largest shareholding shareholder (Prop), corporate growth (Growth). Additionally, the analysis also accounts for the year and industry dummy variables in the empirical process. Table 2 displays the variables and their respective definitions and measurements.

## 5. Empirical results

### 5.1 Description analysis

Table 3 displays the descriptive statistics for the main variables. From Table 3, we know that the mean value of the EID index of private enterprises was 0.134, and 0.147 was the standard deviation. The maximum value was 0.655, which shows that private enterprises' overall EID level was low. The average number of state in private enterprises was 0.019, with a standard deviation of 0.041, showing that state ownership of capital in private companies is low and a maximum shareholding of 0.240, proving that in various companies, the proportion of state-owned participants' shares varies greatly.

### 5.2 Correlation results

Table 4 displays the correlation between the primary research variables. The environmental information disclosure (EID) and the share of state-owned capital invested were positively correlated (coefficient is 0.078, p < 0.01). This indicates that private enterprises disclose environmental information when there is state-owned capital, and the higher the impact on corporate

**Table 2. Definition and measurement of variables.**

| Variable | Symbol | Variable Description |
|---|---|---|
| Environmental information disclosure | EID | Total of all scores of environmental information disclosure index system |
| State-owned capital participation | State | The total shareholding of the state-owned companies among the top 10 shareholders in private companies |
| Senior governance | Govs | Number of state-owned shareholders serving as directors, supervisors, and senior management among the top ten shareholders |
| Media attention | Media | The sum of listed companies' online and paper media reports is added together, and the natural logarithm is taken. |
| Enterprise size | Size | The logarithm of the total number of employees at the company |
| Enterprise age | Age | The logarithm of the observed year minus the year the company was founded |
| Cash flow | Cash | Total assets / Net Cash Flow from Operating Activities |
| The proportion of independent directors | Indep | The ratio of board members to non-employee directors |
| Chairman and CEO concurrently | Dual | The value is 1 if the chairman also acts as CEO; otherwise, the value is 0 |
| Board size | Boards | The number of board members' logarithm |
| Market competition | Com | Sales expenses / gross operating revenue |
| Financial leverage | Lev | Total liabilities / total assets |
| The ratio of the largest shareholding shareholder | Prop | Percentage of largest shareholders among the top ten shareholders |
| Corporate growth | Growth | Operating revenue growth rate |

EID, the higher the shareholder ratio. This preliminarily evidence supports hypothesis $H_1$. Furthermore, there were no significant multicollinearity issues among the key variables.

## 5.3 Regression analysis

Table 5 displays the outcomes of the regression analysis. In column (1), with only year and industry variables as control, and the coefficient for state-owned capital participation was statistically significant at the 1% level. Column (2) further adds control variables to test the effect of state-owned capital involvement (State) on the EID of private firms. The state's regression

**Table 3. Descriptive statistics.**

| Variable | N | Mean | Standard | Minimum | P50 | Maximum |
|---|---|---|---|---|---|---|
| EID | 14339 | 0.134 | 0.147 | 0.000 | 0.069 | 0.655 |
| State | 14339 | 0.019 | 0.041 | 0.000 | 0.000 | 0.240 |
| Govs | 14339 | 0.161 | 0.633 | 0.000 | 0.000 | 9.000 |
| Media | 14399 | 5.564 | 1.130 | 0.000 | 5.624 | 8.216 |
| Size | 14339 | 21.768 | 1.067 | 19.689 | 21.659 | 25.029 |
| Age | 14339 | 16.551 | 5.756 | 4.000 | 16.000 | 31.000 |
| Cash | 14339 | 0.045 | 0.070 | -0.174 | 0.045 | 0.236 |
| Indep | 14339 | 0.378 | 0.052 | 0.333 | 0.364 | 0.571 |
| Dual | 14339 | 0.384 | 0.486 | 0.000 | 0.000 | 1.000 |
| Boards | 14339 | 8.189 | 1.435 | 5.000 | 9.000 | 12.000 |
| Com | 14339 | 0.085 | 0.095 | 0.001 | 0.052 | 0.496 |
| Lev | 14339 | 0.371 | 0.196 | 0.045 | 0.353 | 0.848 |
| Prop | 14339 | 32.142 | 13.650 | 8.380 | 30.080 | 69.280 |
| Growth | 14339 | 0.216 | 0.467 | -0.561 | 0.136 | 3.133 |

**Table 4. Correlation results.**

| Variable | EID | State | Govs | Media | Size | Age | Cash |
|---|---|---|---|---|---|---|---|
| EID | 1 | | | | | | |
| State | 0.078*** | 1 | | | | | |
| Govs | 0.047*** | 0.645*** | 1 | | | | |
| Media | 0.017** | 0.076*** | 0.040*** | 1 | | | |
| Size | 0.299*** | 0.129*** | 0.023*** | 0.294*** | 1 | | |
| Age | 0.130*** | 0.142*** | 0.085*** | -0.003 | 0.200*** | 1 | |
| Cash | 0.157*** | 0.000 | -0.010 | 0.003 | 0.048*** | 0.027*** | 1 |
| Indep | -0.056*** | -0.085*** | -0.093*** | 0.008 | -0.055*** | -0.010 | -0.010 |
| Dual | -0.066*** | -0.060*** | -0.056*** | -0.051*** | -0.121*** | -0.064*** | 0.001 |
| Boards | 0.102*** | 0.140*** | 0.153*** | 0.067*** | 0.168*** | 0.013 | 0.027*** |
| Com | 0.015* | 0.011 | 0.012 | 0.069*** | -0.098*** | -0.006 | 0.119*** |
| Lev | 0.073*** | 0.100*** | 0.040*** | 0.185*** | 0.502*** | 0.208*** | -0.169*** |
| Prop | 0.001 | -0.124*** | -0.118*** | 0.018** | 0.040*** | -0.105*** | 0.083*** |
| Growth | -0.051*** | -0.012 | -0.017** | 0.053*** | 0.077*** | -0.013 | -0.015* |
| Variable | Indep | Dual | Boards | Com | Lev | Prop | Growth |
| Indep | 1 | | | | | | |
| Dual | 0.106*** | 1 | | | | | |
| Boards | -0.635*** | -0.120*** | 1 | | | | |
| Com | 0.016* | 0.046*** | 0.000 | 1 | | | |
| Lev | -0.027*** | -0.089*** | 0.075*** | -0.213*** | 1 | | |
| Prop | 0.033*** | 0.068*** | -0.060*** | 0.011 | -0.021** | 1 | |
| Growth | 0.000 | 0.001 | -0.003 | -0.050*** | 0.075*** | 0.011 | 1 |

Note

*** p<0.01

** p<0.05

* p<0.1.

coefficient was 0.124 at the 1% level of statistical significance. This confirms hypothesis $H_1$, and this finding is consistent with the results of Long et al. [14] who verified the effect of state-owned equity on corporate EID, but the difference is that this study tests the state-owned equity involved in private firms, and it shows that mixed equity structure still promotes EID in private firms. Column (3) tests the moderating impact of media attention on state capital participation and EID; state and media had a 0.042 interaction term coefficient, which was significant at the 5% level, indicating support for hypothesis $H_2$. This finding is consistent with the findings of previous studies [48, 51]. Column (4) tests the moderating effect of appointed directors, supervisors, and executives on state capital participation and environmental information disclosure, at the 5% level, the State and Government interaction term coefficient was significant at 0.034. The findings suggest that the appointment of directors, supervisors and executives can positively moderate the connection between the usage of state-owned capital and the disclosure of environmental information. This also provides empirical evidence of the involvement of state-owned capital in the governance of private enterprises.

## 5.4 Robustness testing

For robustness, we employed several robustness tests. First, the average value (excluding sample enterprises for calculation) of state-owned equity share of private enterprises in the same

**Table 5. Regression test results of the relationship between state owned capital participation and EID of private enterprises.**

| Variable | (1) | (2) | (3) | (4) |
|---|---|---|---|---|
| | EID | EID | EID | EID |
| State | 0.260*** | 0.124*** | -0.142 | 0.008 |
| | (8.26) | (4.07) | (-1.24) | (0.19) |
| Media | | | 0.003** | |
| | | | (2.49) | |
| State*Media | | | 0.042** | |
| | | | (2.26) | |
| Govs | | | | 0.005 |
| | | | | (1.51) |
| State*Govs | | | | 0.034** |
| | | | | (1.98) |
| Size | | 0.039*** | 0.036*** | 0.038*** |
| | | (28.28) | (28.03) | (29.29) |
| Age | | 0.001*** | 0.001*** | 0.001*** |
| | | (3.35) | (3.28) | (3.26) |
| Cash | | 0.052 | 0.050 | 0.050 |
| | | (1.52) | (1.52) | (1.47) |
| Indep | | -0.000 | 0.001 | 0.004 |
| | | (-0.01) | (0.03) | (0.16) |
| Dual | | -0.011*** | -0.011*** | -0.011*** |
| | | (-5.01) | (-5.14) | (-5.10) |
| Boards | | 0.005*** | 0.005*** | 0.005*** |
| | | (4.38) | (4.68) | (4.47) |
| Com | | -0.059*** | -0.063*** | -0.058*** |
| | | (-4.48) | (-4.91) | (-4.51) |
| Lev | | -0.022*** | -0.023*** | -0.022*** |
| | | (-3.08) | (-3.36) | (-3.21) |
| Prop | | -0.000 | -0.000 | -0.000 |
| | | (-0.93) | (-0.78) | (-0.65) |
| Growth | | -0.000*** | -0.000*** | -0.000*** |
| | | (-3.35) | (-3.30) | (-3.29) |
| Constant | 0.035*** | -0.826*** | -0.781*** | -0.803*** |
| | (3.84) | (-25.99) | (-25.85) | (-26.48) |
| Year FE | Yes | Yes | Yes | Yes |
| Industry FE | Yes | Yes | Yes | Yes |
| N | 14339 | 14339 | 14339 | 14339 |
| adj. $R^2$ | 0.177 | 0.256 | 0.262 | 0.262 |

Note

*** $p<0.01$

** $p<0.05$

* $p<0.1$.

industry in the same year was used as an instrumental variable and regressed by two-stage least squares estimation (2SLS). The under-identification test shows that the Anderson LM statistic is 248.87, $p < 0.01$. The estimation results reject the original hypothesis, indicating that the instrumental variable is correlated with the explanatory variables. The weak instrumental

test for the 2SLS shows that the Cragg-Donald Wald F statistic is 252.90, which is greater than the critical value of 10%. Therefore, it is not a weak instrumental variable. Meanwhile, Anderson-Rubin wald also rejected the original hypothesis at the 1% level, indicating a strong correlation between instrumental variables and endogenous variables. The regression results are shown in Table 6, confirming the hypotheses. Second, in order to eliminate selection bias and lessen the effects of the endogeneity problem, we employed the propensity score matching (PSM). We refered to the study of Zhang et al. [44], and set the private firms with state-owned equity holdings of more than 5% among the top 10 shareholders as the treatment group and value = 1. Otherwise, it is 0, which is regarded as the reference group. We used corporate size (Size), corporate age (Age), market competition (Com), cash flow (Cash), gearing ratio (Lev), return on total assets (Roa), percentage of independent directors (Indep), and other variables that may affect state capital participation in private firms. The results in Table 7 show that after the spread of all variables was less than 5% on the standard deviation scale, the balancing hypothesis was confirmed, even if there were substantial differences between the control and experimental groups before the matching process. This finding suggests that after propensity score matching, there was less gap between private enterprises that use state-owned capital and those that do not. Matching the regression results confirms the primary research findings (As shown in Table 8). Finally, in order to exclude the impact of indicator selection on the research results, we replaced the explanatory variable of state-owned capital participation variable. We used the presence or absence of state-owned participating shareholders (Statedummy) among private companies' top ten shareholders firms as a proxy for state-owned capital participation, which is 1 when state-owned participating shareholders are present and 0 otherwise. The regression analysis was then restarted after the main core variables have been replaced. The regression results are shown in Table 8, maintaining the paper's ultimate conclusion.

**Table 6. Instrumental variable results.**

| Variable | First-stage | Second-stage |
|---|---|---|
| | (1) | (2) |
| | State | EID |
| IVState | 0.810*** | |
| | (15.90) | |
| State | | 1.031*** |
| | | (4.89) |
| Constant | -0.114*** | -0.535*** |
| | (-11.69) | (-13.31) |
| Controls | Yes | Yes |
| Year FE | Yes | Yes |
| Industry FE | Yes | Yes |
| N | 14339 | 14339 |
| $R^2$ | 0.080 | 0.089 |
| First stage F value | 62.16 | |
| Cragg-Donald Wald F | 252.90 | |

Note

*** $p<0.01$

** $p<0.05$

* $p<0.1$.

**Table 7. Estimation results and balance test of PSM.**

| Variable | UnMatched Matched | Mean Treated | Control | %bias | %reduct | t-tests t value | p > \|t\| |
|---|---|---|---|---|---|---|---|
| Size | U | 22.029 | 21.736 | 25.2 |  | 10.10 | 0.000 |
|  | M | 22.033 | 22.011 | 1.9 | 92.3 | 0.53 | 0.597 |
| Age | U | 18.726 | 16.303 | 40.8 |  | 15.79 | 0.000 |
|  | M | 18.719 | 18.715 | 0.1 | 99.8 | 0.02 | 0.984 |
| Com | U | 0.089 | 0.085 | 3.9 |  | 1.51 | 0.132 |
|  | M | 0.089 | 0.089 | 0.2 | 94.1 | 0.06 | 0.951 |
| Cash | U | 0.036 | 0.045 | -4.7 |  | -3.02 | 0.003 |
|  | M | 0.042 | 0.043 | -0.4 | 91.1 | -0.30 | 0.766 |
| Lev | U | 0.417 | 0.365 | 25.8 |  | 9.96 | 0.000 |
|  | M | 0.418 | 0.416 | 0.9 | 96.6 | 0.25 | 0.803 |
| Roa | U | 0.105 | 0.044 | 3.2 |  | 2.52 | 0.012 |
|  | M | 0.037 | 0.043 | -0.3 | 90.7 | -1.05 | 0.295 |
| Indep | U | 0.364 | 0.379 | -29.7 |  | -10.46 | 0.000 |
|  | M | 0.364 | 0.364 | 1.5 | 94.8 | 0.48 | 0.628 |

Note: *** $p<0.01$, ** $p<0.05$, * $p<0.1$.

# 6. Extended test

According to the above analysis, state-owned capital introduction into private enterprises can promote the EID of private enterprises to a certain extent. This research argues that EID might be greatly bolstered by including state-owned capital in privately-held businesses, increasing their profits. Therefore, this paper uses profitability (Roa) as the intermediary variable to further analyze this mechanism. Based on model (1), regression model (4) and model (5) were constructed to further examine the mediating role of profitability on state-owned capital

**Table 8. Results of the robustness test.**

| Variable | (1) EID | (2) EID |
|---|---|---|
| State | 0.115*** |  |
|  | (3.31) |  |
| Statedummy |  | 0.004* |
|  |  | (1.90) |
| Constant | -0.917*** | -0.828*** |
|  | (-16.61) | (-25.87) |
| Controls | Yes | Yes |
| Year FE | Yes | Yes |
| Industry FE | Yes | Yes |
| N | 4240 | 14339 |
| adj. $R^2$ | 0.261 | 0.254 |

Note
*** $p<0.01$
** $p<0.05$
* $p<0.1$.

participation and environmental information disclosure of private enterprises:

$$Roa_{it} = \beta_0 + \beta_1 State_{it} + \sum Control-Variables + Year + Industry + \varepsilon_{it} \qquad (4)$$

$$EID_{it} = \beta_0 + \beta_1 State_{it} + \beta_2 Roa_{it} + \sum Control-Variables + Year + Industry + \varepsilon_{it} \qquad (5)$$

This study investigated how profitable businesses are estimated by calculating return on assets (net profit divided by total assets). Shown in Table 9 are the findings of the regression analysis Column (1). State-owned capital participation is associated with profitability in a way that is statistically significant at 1%, as shown by the link between State and Roa, which is 0.027;

**Table 9. Intermediary effect test.**

| Variable | (1) | (2) |
|---|---|---|
|  | Roa | EID |
| State | 0.398** | 0.116*** |
|  | (2.09) | (3.79) |
| Roa |  | 0.013*** |
|  |  | (10.08) |
| Size | 0.051*** | 0.037*** |
|  | (2.63) | (29.06) |
| Age | 0.003** | 0.001*** |
|  | (2.09) | (3.07) |
| Cash | -5.983** | 0.127*** |
|  | (-2.27) | (9.60) |
| Indep | 0.076 | 0.006 |
|  | (0.72) | (0.22) |
| Dual | 0.001 | -0.011*** |
|  | (0.08) | (-5.12) |
| Boards | 0.011* | 0.005*** |
|  | (1.92) | (4.69) |
| Com | 0.134* | -0.059*** |
|  | (1.72) | (-4.62) |
| Lev | -0.628*** | -0.014** |
|  | (-2.85) | (-2.21) |
| Prop | 0.003*** | -0.000 |
|  | (2.82) | (-1.22) |
| Growth | -0.000 | -0.000*** |
|  | (-0.29) | (-3.46) |
| Constant | -0.889** | -0.788*** |
|  | (-2.49) | (-26.20) |
| Year FE | Yes | Yes |
| Industry FE | Yes | Yes |
| N | 14339 | 14339 |
| adj. $R^2$ | 0.548 | 0.264 |

Note

*** p<0.01

** p<0.05

* p<0.1.

Column (2) results in a coefficient of 3.411 that was statistically significant (at the 1% level) for corporate Roa on EID, and a similarly significant regression coefficient for the state, suggesting that the relationship between governmental ownership of capital and the EID of private enterprises is somewhat mediated by corporate profitability.

## 7. Conclusion

### 7.1 Discussion

This study is pivotal in the context of China's green economic transformation and improvement, emphasizing the need for high-quality economic development, reduced environmental pollution, and promoting private enterprises to actively assume environmental responsibility. Based on data from 2010–2020, and a selection of private companies, the paper examines how mixed-reform private firms impact environmental information disclosure. The results show that corporate EID can be greatly boosted through the use of state-owned capital in private businesses. A higher percentage of state-owned equity corresponds to a more substantial positive impact on environmental transparency within corporate practices.

While some studies have explored how an organization's equity structure and governance characteristics affect corporate EID [12, 30, 52, 53]. For example, Shaheen et al. [52] found that female participation significantly contributes to corporate social disclosure. Fernandes et al. [30] found that the characteristics of the board of directors influence the level of EID. We specifically distinguished this equity structure with a mix of state-owned and private firms, extending the previous literature on the impact of corporate governance on EID. We find that, firstly, the introduction of state-owned capital by private firms enhances their ability to access resources and effectively alleviates financing constraints, which helps to improve environmental performance. At the same time, being subject to more supervision and checks and balances, private firms will be more active in disclosing environmental information in order to gain political legitimacy. Second, the nomination of directors, supervisors, and executives by state-owned shareholders may positively moderate the EID between state-owned capital involvement and private enterprises. When private enterprises receive stronger media attention more than any other factor, the presence of a company's capital structure that includes state-owned capital positively affects the latter's EID. This work complements previous research on external pressure [10] and shareholder governance [12]. Finally, we find that profitability partially mediates the effect of state ownership on the EID of private enterprises. Although profitability has always been considered an important factor affecting EID, empirical results are different. Dyduch and Krasodomska [54] and Yin et al. [55] argue that the impact of profitability is inconclusive. Our research results suggest that state-owned capital can positively impact the environmental information disclosure of private enterprises by enhancing their profitability. In summary, this study underlines the pivotal role of state-owned capital in promoting EID within private enterprises, contributing to China's sustainable and environmentally responsible economic development.

### 7.2 Practical implications

The findings of this study offer practical insights into China's growing private sector and mixed-ownership economy to raise the level of EID. First, increasing the degree of EID by private enterprises can be accomplished by involving state-owned capital in those firms. To maintain and expand the mixed-ownership reform, the promotion of state-owned capital through equity investment merger and reorganization, and other ways to integrate resources with private enterprises. By maximizing the utilization of state-owned capital, private enterprises can be motivated to actively disclose environmental information. Second, giving full play to the

supervisory and governance influence of state-owned capital and fostering more enthusiasm among using state-owned capital for corporate governance, special emphasis should be placed on the governance participation of state-owned equity, such as the selection of state-owned directors, the improvement of their governance structures, and the optimization of current decision-making processes. Finally, strengthening external oversight and governance is essential. Media coverage has the potential to encourage private businesses to disclose environmental information under public pressure. Therefore, to perform the function of public opinion polling more effectively, the media should report the environmental performance of enterprises more actively, encouraging firms to improve their environmental practices and disclose environmental information more comprehensively.

### 7.3 Limitations and future research directions

While this study provides valuable insights, it has certain limitations, such as the number of companies in the sample, the selection of indicators for variables, and the method of statistical data. First, due to the complexity of collecting and organizing reports manually, we only investigated listed companies in Shenzhen and Shanghai shares in China. Future research could aim to expand the sample size. Second, the content analysis method used for EID evaluation is subjective to some extent. In the era of big data, employing cutting-edge methods to assess EID and differentiating between research on "soft" disclosures (environmental governance policies, environmental concepts, goals, etc.) and "hard" disclosures (pollution emissions, pollution fines, governance investments, etc.) could enhance the research in this area. These limitations present opportunities for future research to build upon the findings of this study and explore new dimensions of EID and corporate governance in mixed-ownership settings.

## Author Contributions

**Conceptualization:** Aihua Xiong.

**Data curation:** Tingting Song.

**Formal analysis:** Tingting Song.

**Investigation:** Tingting Song, Aihua Xiong.

**Resources:** Aihua Xiong.

**Supervision:** Aihua Xiong.

**Visualization:** Tingting Song, Aihua Xiong.

**Writing – original draft:** Tingting Song.

**Writing – review & editing:** Aihua Xiong.

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
