## [Decision Letter · Decision Letter 0]

24 Sep 2023

PONE-D-23-25299Environmental Disclosure Practices in Mixed Ownership Models: A Study of Chinese Private EnterprisesPLOS ONE

Dear Dr. Song,

Thank you for submitting your manuscript to PLOS ONE. After careful consideration, we feel that it has merit but does not fully meet PLOS ONE’s publication criteria as it currently stands. Therefore, we invite you to submit a revised version of the manuscript that addresses the points raised during the review process.

We look forward to receiving your revised manuscript.

Kind regards,

Khanh Hoang, Ph.D.

Academic Editor

PLOS ONE

Journal Requirements:

2. Thank you for submitting the above manuscript to PLOS ONE. During our internal evaluation of the manuscript, we found significant text overlap between your submission and previous work in the [introduction, conclusion, etc.].

Please revise the manuscript to rephrase the duplicated text, cite your sources, and provide details as to how the current manuscript advances on previous work. Please note that further consideration is dependent on the submission of a manuscript that addresses these concerns about the overlap in text with published work.

[If the overlap is with the authors’ own works: Moreover, upon submission, authors must confirm that the manuscript, or any related manuscript, is not currently under consideration or accepted elsewhere. If related work has been submitted to PLOS ONE or elsewhere, authors must include a copy with the submitted article. Reviewers will be asked to comment on the overlap between related submissions (http://journals.plos.org/plosone/s/submission-guidelines#loc-related-manuscripts).]

We will carefully review your manuscript upon resubmission and further consideration of the manuscript is dependent on the text overlap being addressed in full. Please ensure that your revision is thorough as failure to address the concerns to our satisfaction may result in your submission not being considered further.

Additional Editor Comments:

I have completed the evaluation of the manuscript. Based on my own reading of the manuscript and the comments from reviewers, I suggest a major revision of the paper. Please closely follow the comments and suggestions from the reviewers, make clear point-to-point responses and proofread the paper carefully.

Reviewers' comments:

Reviewer's Responses to Questions

**Comments to the Author**

1. Is the manuscript technically sound, and do the data support the conclusions?

Reviewer #1: Partly

Reviewer #2: Yes

2. Has the statistical analysis been performed appropriately and rigorously? 

Reviewer #1: Yes

Reviewer #2: Yes

3. Have the authors made all data underlying the findings in their manuscript fully available?

Reviewer #1: Yes

Reviewer #2: No

4. Is the manuscript presented in an intelligible fashion and written in standard English?

Reviewer #1: Yes

Reviewer #2: Yes

5. Review Comments to the Author

Reviewer #1: REVIEW RESULT

Manuscript_PONE-D-23-25299t: Environmental Disclosure Practices in Mixed Ownership Models: A Study of Chinese Private Enterprises

1. OBJECTIVE: The objective of the article is met with the paper’s scope.

2. INTRODUTION: It is important to provide clarity regarding your research questions, objectives, the motivation behind your study, both in terms of background and theory, as well as the empirical motivation. Additionally, you should outline how your study contributes to the existing literature. This can be achieved by explicitly stating your research questions and objectives, identifying potential theoretical and background motivations or gaps, and elucidating how your research adds to the body of knowledge in this field. Authors need to elaborate on the novel contribution of this study compared to previous research, shedding light on why the topic of "Environmental Disclosure Practices in Mixed Ownership Models" was chosen, and how the choice of China as the context differs from other studies addressing the same topic within a similar context. Please see the following papers:

1. Kilincarslan, E., Elmagrhi, M.H. and Li, Z. (2020), "Impact of governance structures on environmental disclosures in the Middle East and Africa", Corporate Governance, Vol. 20 No. 4, pp. 739-763. https://doi.org/10.1108/CG-08-2019-0250

2. Bin, L., Lei, X., Ron, P. M., Xin, L., and Ailing, Pan. (2022), “Mixed-Ownership Reform and Private Firms’ Corporate Social Responsibility Practices: Evidence From China, Business & Society, Vol. 61(2) 389–418. http://doi.org/10.1177/0007650320958762

3. Meng, X.H., Zeng, S.X., Jonathan, J. S., Qi, G.Y. and Zhang. Z.B. (2014), "The relationship between corporate environmental performance and environmental disclosure: An empirical study in China", Journal of Environmental Management, Vol. 145, p.357-367, https://doi.org/10.1016/j.jenvman.2014.07.009.

3. LITERATURE REVIEW: PRIOR STUDIES AND HYPOTHESES DEVELOPMENT

- COMPREHENSIVE PRIOR LITERATURE: There are appropriate and adequate references to related and previous research. Literature review and hypothesis development should be discussed in more detail, in my opinion. Please offer a comprehensive theoretical framework that will explain the underlying assumptions and hypotheses of interest. Please use both seminal (old) and recently (newly) published studies to support your argument, making sure to clearly state how they contribute to the connection between the dependent and independent variables.

Review of the literature and development of hypotheses: Please strengthen your hypotheses drawing from theory, (ii) referencing empirical research, (iii) considering the study setting and context, and (iv) finally putting them into place. You should do this by referencing both classic (previously published) and contemporary (recently published) studies. Most of the recent related studies mentioned in your study are just in 2020 and earlier. It should be updated.

- HYPOTHESIS DEVELOPMENT: The hypotheses have been developed based on previous studies.

4. METHODOLOGY, RESEARCH DESIGN AND METHODS

- The research methodology in this study is sound.

- Please explain why the authors chose to calculate the maximum score of 37 points for companies to determine their EID and then divide the EID scores of sample firms by this maximum score to obtain the company's EID index. The authors should reference reputable studies that have employed this method to measure EID scores.

- Further evidence is needed to support the argument that there is a lag in the effects of participation by state-owned capital on private firms' EID, suggesting that there is no immediate effect in the year of participation. This would enhance the persuasiveness of using lag variables when conducting Robustness tests.

5. FINDINGS & DISCUSSION

- In the research section, it is important for the authors to provide clear conclusions regarding the hypotheses and indicate which hypotheses have been accepted or rejected. Furthermore, the authors should discuss how their results align with prior studies or use relevant theories to explain the observed outcomes.

- Below each result table, there should be additional Notes regarding the applied regression method, dependent variables, independent variables, and the significance symbols *, **, and *** in the result tables.

- Table 3 requires a more detailed description. The authors should provide a more in-depth analysis of the significance of the minimum, maximum, mean, and standard deviation values of the dependent and independent variables in the model.

- Specify the Multivariate regression methods more explicitly in Table 5.

6. ENGLISH LANGUAGE AND STYLE:

It has some minor mistakes. The article could be proofread by a native speaker to enhance the quality of communication.

7. OTHER SUGGESTIONS FOR THE AUTHOR(S):

- Keywords should be arranged in alphabetical order.

- The list of references should adhere to the required citation style. For example:

1. Syed AM, Ntim CG. Environment, social, and governance (ESG) criteria and preference of managers. Cogent Bus Manage. 2017; 4:1340820. https://doi.org/10.1080/23311975.2017.1340820.

2. Boonlert-U-Thai K, Meek JK, Nabar S. Earnings attributes and investor-protection: international evidence. Int J Account. 2006; 41:327–57. https://doi.org/10.1016/J.INTACC.2006.09.008.

Reviewer #2: The research topic appears relevant and timely, given increasing concerns about environmental sustainability and corporate transparency.

The paper's overall structure is sound, with logical transitions between sections. However, some sentences are quite lengthy and complex.

It is quite hard for reader to understand about the calculation of EID (EID = 100* ( EIDI/39)). why using EIDI divide 39?

And this measurement you create or take advance from prior research? Please explain more and citation (if have).

In the part Explanatory variables- State-owned capital participation (State), you mentioned "We do this by consulting prior research" but no reference cited here. Please provide reference.

It will be better when you explain more about the Moderating variables- Media attention and the methodology applied (Python programming).

6. PLOS authors have the option to publish the peer review history of their article (what does this mean?). If published, this will include your full peer review and any attached files.

Reviewer #1: No

Reviewer #2: No

---

## [Author Response · Author response to Decision Letter 0]

11 Oct 2023

Dear editor and reviewer:

Hello everyone, thank you very much to the editors and reviewers for carefully reviewing this article and providing valuable suggestions for revisions. I have made careful revisions and supplements according to the reviewer's comments, and all modifications have been highlighted in red font in the revised manuscript.

I have made revisions and supplements to the attached modification comments item by item.

Please refer to the attachment (Response to Reviewers) for the details of the one-on-one response.

---

## [Decision Letter · Decision Letter 1]

13 Nov 2023

PONE-D-23-25299R1Environmental Disclosure Practices in Mixed Ownership Models: A Study of Chinese Private EnterprisesPLOS ONE

Dear Dr. Song,

Thank you for submitting your manuscript to PLOS ONE. After careful consideration, we feel that it has merit but does not fully meet PLOS ONE’s publication criteria as it currently stands. Therefore, we invite you to submit a revised version of the manuscript that addresses the points raised during the review process.

We look forward to receiving your revised manuscript.

Kind regards,

Khanh Hoang, Ph.D.

Academic Editor

PLOS ONE

Additional Editor Comments:

I am writing to inform the decision on the manuscript number PONE-D-23-25299R1 "Environmental Disclosure Practices in Mixed Ownership Models: A Study of Chinese Private Enterprises". The author has addressed all comments from the two reviewers. However, when I read the revised manuscript one more time, I see that there are more parts of the manuscript need to be improved before I can suggest acceptance of the submission. I list them as follows:

1. Instrumental variable (IV) approach: The author uses "average value of state-owned equity share of private enterprises in the same industry in the same year" as the IV. At this current form, the proposed IV cannot pass the exclusion restrictions of IV/2SLS. I suggest the author exclude the current firm before calculating the cross-sectional industry average of state ownership. That way is a trick to unlink the current firm's relatedness to that IV, thus rendering it more exogenous. Also, the author must report the under-identification, weak-identification test statistics, and the Anderson-Rubin confidence interval for the IV/2SLS estimation in Table 6.

2. Propensity Score Matching (PSM) approach: Please report the matching diagnostics (balance diagnostics) for the pre-match and post-match samples. Also, what happened if the state ownership is very small, like less than 5%? It is not significant to be considered a major shareholder. How do you clarify this?

3. Why using two different explanatory variables ("State" & "State_")? The results of the two regressions in this table therefore cannot be comparable.

4. In tables: please clearly indicate the fixed effects by replace "Industry" with "Industry fixed effect", similar to "Year" changing to "Year fixed effect".

Once the author addresses those comments, the manuscript then can be ready for publication.

Reviewers' comments:

Reviewer's Responses to Questions

**Comments to the Author**

1. If the authors have adequately addressed your comments raised in a previous round of review and you feel that this manuscript is now acceptable for publication, you may indicate that here to bypass the “Comments to the Author” section, enter your conflict of interest statement in the “Confidential to Editor” section, and submit your "Accept" recommendation.

Reviewer #1: All comments have been addressed

Reviewer #2: All comments have been addressed

2. Is the manuscript technically sound, and do the data support the conclusions?

Reviewer #1: Yes

Reviewer #2: Yes

3. Has the statistical analysis been performed appropriately and rigorously? 

Reviewer #1: Yes

Reviewer #2: Yes

4. Have the authors made all data underlying the findings in their manuscript fully available?

Reviewer #1: Yes

Reviewer #2: Yes

5. Is the manuscript presented in an intelligible fashion and written in standard English?

Reviewer #1: Yes

Reviewer #2: Yes

6. Review Comments to the Author

Reviewer #1: REVIEW RESULT

Manuscript_PONE-D-23-25299R1: Environmental Disclosure Practices in Mixed Ownership Models: A Study of Chinese Private Enterprises

Reviewer: Dr. Le Quynh Lien

1. OBJECTIVE: The objective of the article is met with the paper’s scope.

2. INTRODUTION: Authors added how this study contributes to the existing literature and elaborate on the novel contribution of this study compared to previous research.

3. LITERATURE REVIEW: PRIOR STUDIES AND HYPOTHESES DEVELOPMENT

- COMPREHENSIVE PRIOR LITERATURE: There are appropriate and adequate references to related and previous research. Literature review and hypothesis development discussed in more detail, in my opinion.

- HYPOTHESIS DEVELOPMENT: The hypotheses have been developed based on previous studies.

4. METHODOLOGY, RESEARCH DESIGN AND METHODS

- The research methodology in this study is sound.

- Authors explained why they chose to calculate the maximum score of 37 points for companies to determine their EID and then divide the EID scores of sample firms by this maximum score to obtain the company's EID index.

- They enhanced the persuasiveness of using lag variables when conducting Robustness tests.

5. FINDINGS & DISCUSSION

- They provided clearer conclusions about the assumptions and discussed their consistency with previous research.

6. ENGLISH LANGUAGE AND STYLE:

Keywords arranged in alphabetical order. The references of the article have been modified as required.

Reviewer #2: Thank you for addressing all my comments. Overall, your study after revised is sound. It provides significantly contributes to understanding how state-owned capital participation affects environmental information disclosure in private enterprises.

7. PLOS authors have the option to publish the peer review history of their article (what does this mean?). If published, this will include your full peer review and any attached files.

Reviewer #1: **Yes: **Quynh Lien Le, National Economics University.

Reviewer #2: No

---

## [Author Response · Author response to Decision Letter 1]

15 Nov 2023

Response to Reviewers

Dear editor and reviewer:

Hello everyone, thank you very much to the editors and reviewers for carefully reviewing this article and providing valuable suggestions for revisions. I have made careful revisions and supplements according to the reviewer's comments, and all modifications have been highlighted in red font in the revised manuscript.

I have made revisions and supplements to the attached modification comments item by item, as follows:

Additional Editor Comments, Concern # 1: Instrumental variable (IV) approach: The author uses "average value of state-owned equity share of private enterprises in the same industry in the same year" as the IV. At this current form, the proposed IV cannot pass the exclusion restrictions of IV/2SLS. I suggest the author exclude the current firm before calculating the cross-sectional industry average of state ownership. That way is a trick to unlink the current firm's relatedness to that IV, thus rendering it more exogenous. Also, the author must report the under-identification, weak-identification test statistics, and the Anderson-Rubin confidence interval for the IV/2SLS estimation in Table 6.

Author response: Based on the reviewer's suggestion, I excluded the current firm before calculating the cross sectional industry average of state ownership. Then recalculated and updated Table 6 and related content, reporting the under identification, weak identification test statistics, and the Anderson Rubin confidence interval for the IV/2SLS estimation. 

Author action: The following is a modified portion of the paper in response to the comments：

 The average value (excluding sample enterprises for calculation) of state-owned equity share of private enterprises in the same industry in the same year was used as an instrumental variable and regressed by two-stage least squares estimation (2SLS). The under-identification test shows that the Anderson LM statistic is 248.87, p < 0.01. The estimation results reject the original hypothesis, indicating that the instrumental variable is correlated with the explanatory variables.The weak instrumental test for the 2SLS shows that the Cragg-Donald Wald F statistic is 252.90, which is greater than the critical value of 10%. Therefore, it is not a weak instrumental variable. Meanwhile, Anderson-Rubin wald also rejected the original hypothesis at the 1% level, indicating a strong correlation between instrumental variables and endogenous variables. 

Table 6. Instrumental variable results.

Variable First-stage Second-stage

 (1) (2)

 State EID

IVState 0.810*** 

 (15.90) 

State 1.031***

 (4.89)

Constant -0.114*** -0.535***

 (-11.69) (-13.31)

Controls Yes Yes

Year FE Yes Yes

Industry FE Yes Yes

N 14339 14339

R2 0.080 0.089

First stage F value 62.16 

Cragg-Donald Wald F 252.90 

Note: *** p<0.01, ** p<0.05, * p<0.1.

Additional Editor Comments, Concern # 2: Propensity Score Matching (PSM) approach: Please report the matching diagnostics (balance diagnostics) for the pre-match and post-match samples. Also, what happened if the state ownership is very small, like less than 5%? It is not significant to be considered a major shareholder. How do you clarify this?

Author response: Based on the reviewer's suggestion, I reported the matching diagnostics (balance diagnostics) for the pre-match and post-match samples. At the same time, I have carefully considered the possibility that state-owned equity may be very small, which cannot reflect this issue. I believe this issue is indeed worth my deep consideration. Through a review of existing literature, most scholars measure state-owned capital participation based on the proportion of China's equity held by the top ten shareholders of private enterprises. I believe another reason may be the issue of data acquisition. Obtaining all shareholder information in a company may be difficult, and I hope to solve this problem in future research. At the same time, refer to the research of Scholar Zhang. 

Author action: The following is a modified portion of the paper in response to the comments：

We refered to the study of Zhang et al. [45], and set the private firms with state-owned equity holdings of more than 5% among the top 10 shareholders as the treatment group and value = 1. Otherwise, it is 0, which is regarded as the reference group. We used corporate size (Size), corporate age (Age), market competition (Com), cash flow (Cash), gearing ratio (Lev), return on total assets (Roa), percentage of independent directors (Indep), and other variables that may affect state capital participation in private firms. The results in Table 7 show that after the spread of all variables was less than 5% on the standard deviation scale, the balancing hypothesis was confirmed, even if there were substantial differences between the control and experimental groups before the matching process. This finding suggests that after propensity score matching, there was less gap between private enterprises that use state-owned capital and those that do not. Matching the regression results confirms the primary research findings (As shown in Table 8).

Table 7. Estimation results and balance test of PSM

 UnMatched Mean t-tests

Variable Matched Treated Control %bias %reduct t value p > |t|

Size U 22.029 21.736 25.2 10.10 0.000

 M 22.033 22.011 1.9 92.3 0.53 0.597

Age U 18.726 16.303 40.8 15.79 0.000

 M 18.719 18.715 0.1 99.8 0.02 0.984

Com U 0.089 0.085 3.9 1.51 0.132

 M 0.089 0.089 0.2 94.1 0.06 0.951

Cash U 0.036 0.045 -4.7 -3.02 0.003

 M 0.042 0.043 -0.4 91.1 -0.30 0.766

Lev U 0.417 0.365 25.8 9.96 0.000

 M 0.418 0.416 0.9 96.6 0.25 0.803

Roa U 0.105 0.044 3.2 2.52 0.012

 M 0.037 0.043 -0.3 90.7 -1.05 0.295

Indep U 0.364 0.379 -29.7 -10.46 0.000

 M 0.364 0.364 1.5 94.8 0.48 0.628

Note: *** p<0.01, ** p<0.05, * p<0.1.

Table 8. Results of the robustness test.

Variable (1) (2)

 EID EID

State 0.115*** 

 (3.31) 

Statedummy 0.004*

 (1.90)

Constant -0.917*** -0.828***

 (-16.61) (-25.87)

Controls Yes Yes

Year FE Yes Yes

Industry FE Yes Yes

N 4240 14339

adj. R2 0.261 0.254

Note: *** p<0.01, ** p<0.05, * p<0.1.

Additional Editor Comments, Concern # 3:Why using two different explanatory variables ("State" & "State_")? The results of the two regressions in this table therefore cannot be comparable.

Author response: This article also uses the method of replacing explanatory variables to test robustness, where State_ The explanatory variable used to represent substitution may have some issues with the wording in the text. Therefore, I changed to Statedummy to represent the replaced explanatory variable and modified some of the textual content to make the logic clearer.

Author action: Finally, in order to exclude the impact of indicator selection on the research results, we replaced the explanatory variable of state-owned capital participation variable. We used the presence or absence of state-owned participating shareholders (Statedummy) among private companies' top ten shareholders firms as a proxy for state-owned capital participation, which is 1 when state-owned participating shareholders are present and 0 otherwise.The regression analysis was then restarted after the main core variables have been replaced. The regression results are shown in Table 8, maintaining the paper's ultimate conclusion.

Additional Editor Comments, Concern # 4: In tables: please clearly indicate the fixed effects by replace "Industry" with "Industry fixed effect", similar to "Year" changing to "Year fixed effect".

Author response: According to the reviewer's suggestion, I have changed the “Industry” and “Year” in each table to “Industry FE” and “Year FE”.

Reviewer#1, 

1. OBJECTIVE: The objective of the article is met with the paper’s scope. 

2. INTRODUTION: Authors added how this study contributes to the existing literature and elaborate on the novel contribution of this study compared to previous research.

3. LITERATURE REVIEW: PRIOR STUDIES AND HYPOTHESES DEVELOPMENT

- COMPREHENSIVE PRIOR LITERATURE: There are appropriate and adequate references to related and previous research. Literature review and hypothesis development discussed in more detail, in my opinion. 

- HYPOTHESIS DEVELOPMENT: The hypotheses have been developed based on previous studies. 

4. METHODOLOGY, RESEARCH DESIGN AND METHODS

- The research methodology in this study is sound.

- Authors explained why they chose to calculate the maximum score of 37 points for companies to determine their EID and then divide the EID scores of sample firms by this maximum score to obtain the company's EID index. 

- They enhanced the persuasiveness of using lag variables when conducting Robustness tests.

5. FINDINGS & DISCUSSION

- They provided clearer conclusions about the assumptions and discussed their consistency with previous research.

6. ENGLISH LANGUAGE AND STYLE:

Keywords arranged in alphabetical order. The references of the article have been modified as required.

Author response: Thank you to the reviewer for accepting my revised article.

Reviewer#2,

Thank you for addressing all my comments. Overall, your study after revised is sound. It provides significantly contributes to understanding how state-owned capital participation affects environmental information disclosure in private enterprises.

Author response: Thank you to the reviewer for accepting my revised article.

Tingting Song

15/11/2023

---

## [Editor Report · Decision Letter 2]

21 Nov 2023

Environmental Disclosure Practices in Mixed Ownership Models: A Study of Chinese Private Enterprises

PONE-D-23-25299R2

Dear Dr. Song,

We’re pleased to inform you that your manuscript has been judged scientifically suitable for publication and will be formally accepted for publication once it meets all outstanding technical requirements.

Kind regards,

Khanh Hoang, Ph.D.

Academic Editor

PLOS ONE

---

## [Editor Report · Acceptance letter]

23 Nov 2023

PONE-D-23-25299R2 

Environmental disclosure practices in mixed ownership models: A study of Chinese private enterprises 

Dear Dr. Song:

I'm pleased to inform you that your manuscript has been deemed suitable for publication in PLOS ONE. Congratulations! Your manuscript is now with our production department. 

Kind regards, 

on behalf of

Dr Khanh Hoang 

Academic Editor

PLOS ONE